# Whole-Genome Sequencing of *Staphylococcus aureus* and *Staphylococcus haemolyticus* Clinical Isolates from Egypt

Cesar Montelongo,[a] Carine R. Mores,[a] Catherine Putonti,[a,b,c] Alan J. Wolfe,[a] Alaa Abouelfetouh[d,e]

[a]Department of Microbiology and Immunology, Stritch School of Medicine, Loyola University Chicago, Maywood, Illinois, USA
[b]Bioinformatics Program, Loyola University Chicago, Chicago, Illinois, USA
[c]Department of Biology, Loyola University Chicago, Chicago, Illinois, USA
[d]Department of Microbiology and Immunology, Faculty of Pharmacy, Alexandria University, Alexandria, Egypt
[e]Department of Microbiology and Immunology, Faculty of Pharmacy, Alamein International University, Alamein, Egypt

**ABSTRACT**  Infections caused by antibiotic-resistant *Staphylococcus* are a global concern. This is true in the Middle East, where increasingly resistant *Staphylococcus aureus* and *Staphylococcus haemolyticus* strains have been detected. While extensive surveys have revealed the prevalence of infections caused by antibiotic-resistant staphylococci in Europe, Asia, and North America, the population structure of antibiotic-resistant staphylococci recovered from patients and clinical settings in Egypt remains uncharacterized. We performed whole-genome sequencing of 56 *S. aureus* and 10 *S. haemolyticus* isolates from Alexandria Main University Hospital; 46 of the *S. aureus* genomes and all 10 of the *S. haemolyticus* genomes carry *mecA*, which confers methicillin resistance. Supplemented with additional publicly available genomes from the other parts of the Middle East (34 *S. aureus* and 6 *S. haemolyticus*), we present the largest genomic study to date of staphylococcal isolates from the Middle East. These genomes include 20 *S. aureus* multilocus sequence types (MLST), including 3 new ones. They also include 9 *S. haemolyticus* MLSTs, including 1 new one. Phylogenomic analyses of each species' core genome largely mirrored those of the MLSTs, irrespective of geographical origin. The hospital-acquired *spa* t037/ST239-SCC*mec* III/MLST CC8 clone represented the largest clade, comprising 22% of the *S. aureus* isolates. Like *S. aureus* genome surveys of other regions, these isolates from the Middle East have an open pangenome, a strong indicator of gene exchange of virulence factors and antibiotic resistance genes with other reservoirs. Our genome analyses will inform antibiotic stewardship and infection control plans in the Middle East.

**IMPORTANCE**  Staphylococci are understudied despite their prevalence within the Middle East. Methicillin-resistant *Staphylococcus aureus* (MRSA) is endemic to hospitals in Egypt, as are other antibiotic-resistant strains of *S. aureus* and *S. haemolyticus*. To provide insight into the strains circulating in Egypt, we performed whole-genome sequencing of 56 *S. aureus* and 10 *S. haemolyticus* isolates from Alexandria Main University Hospital. Through analysis of these genomes, as well as all available *S. aureus* and *S. haemolyticus* genomes from the Middle East ($n = 40$), we were able to produce a picture of the diversity in this region more complete than those afforded by traditional molecular typing strategies. For example, we identified 4 new MLSTs. Most strains harbored genes associated with multidrug resistance, toxin production, biofilm formation, and immune evasion. These data provide invaluable insight for future antibiotic stewardship and infection control within the Middle East.

**KEYWORDS** *Staphylococcus aureus*, *Staphylococcus haemolyticus*, Middle East, MLST

Address correspondence to Alaa Abouelfetouh, Alaa.abouelfetouh@pharmacy.alexu.edu.eg.

The authors declare a conflict of interest. A.J.W. is a member of the Advisory Board of Urobiome Therapeutics. The remaining authors report no disclosures.

Staphylococci are a heterogenous group of commensal bacteria in humans with the potential to cause infections (1). Two staphylococcal species especially relevant to the clinical setting are *Staphylococcus aureus* and *Staphylococcus haemolyticus*. *S. aureus* is

arguably the most clinically important staphylococcal species; it can cause mild erythema or serious life-threatening ailments, including septicemia, pneumonia, and endocarditis (2). A challenge in treating and controlling *S. aureus* stems from both its prevalence and its increasing resistance to clinically used antibiotics. Together, these factors make *S. aureus* one of the leading agents of nosocomial and community-acquired infections (3, 4). *S. haemolyticus* is the second most common staphylococcal species isolated from human blood culture. It can be a reservoir for antibiotic resistance genes, which can be shared with other staphylococci, including *S. aureus* (5–7).

Epidemiological surveillance and profiling are key to managing staphylococci (8, 9). Historically, profiling of staphylococci has relied on complementary molecular typing strategies (10). Multilocus sequence typing (MLST) is effective at tracking a broad range of clones (10). *spa* typing complements MLST by tracking the molecular evolution of *S. aureus*, given the relevance of protein A to the infectious process (9). Subtyping elements in the staphylococcal cassette chromosome *mec* (SCC*mec*) profiles clinically relevant antibiotic resistances, including *mecA*, associated with methicillin-resistant *S. aureus* (MRSA) (11, 12). Lastly, *S. aureus* strains are often assayed for the virulence factor Panton-Valentine leucocidin (PVL), which is common among community-acquired MRSA (CA-MRSA) strains and rare among hospital-associated MRSA (HA-MRSA) strains (11). PVL is thought to contribute to epidemic spread (13), and many MRSA strains in circulation in the United States and Europe, e.g., USA300 strains, are PVL-positive (14). Together, these profiling strategies can be a powerful means to type, trace, and manage staphylococcal infections, but technical limitations curtail the usefulness of molecular typing in real time (9, 10). In contrast, studies have demonstrated that whole-genome sequencing (WGS) can be used to type, discriminate, and cluster staphylococcal isolates for the purpose of outbreak control (15–17). WGS can identify outbreak clones or clades, groups of independent isolates that share phenotypic and genotypic traits, most likely have a common ancestor, and form a branch on a phylogenetic tree (18–20). WGS could be used to close the gap in staphylococcal management in regions that have not been extensively monitored, such as the Middle East and specifically Egypt.

In contrast to that in Europe, Asia, and the United States, where epidemiological and typing data are abundant (21), the epidemiology of staphylococci in non-European countries of the Mediterranean region or the Arab world is understudied (22, 23). This is problematic, as antibiotic resistance in *S. haemolyticus* has been detected in the Middle East, in countries such as Turkey (6) and Egypt (7). Also, MRSA is prevalent in and endemic to hospitals in this region, with a median MRSA prevalence of 38% in Algeria, Cyprus, Egypt, Jordan, Lebanon, Malta, Morocco, Tunisia, and Turkey (24). A recent prospective study found the prevalence of MRSA among *S. aureus* infections to be as high as 67% in the Levant (Lebanon, Palestine, Jordan, and Iraq) and 57% in Egypt, Algeria, and Tunisia (25). Another recent study of hospital health care workers in Oman found that >20% carry *S. aureus* in their nose, with 63.4% being MRSA strains (26). PVL prevalence is reported to be low in some of these countries, indicating HA-MRSA predominance (22, 27, 28). PVL-positive strains are most frequently associated with skin and soft tissue disease, although they can be associated with pneumonia and bacteremia (14). Research into the epidemiology of the staphylococci in the Middle East is urgent.

Molecular typing and phylogenetic data from the Middle East are limited to just a few studies. Multiple isolates from Palestine were typed as sequence type 22 (ST-22), with a minority typed as ST-80-MRSA-IV and PVL-positive (29). The majority of *S. aureus* isolates from Jordan were ST-80-MRSA-IV (30). In Lebanon, the primary lineage was PVL-positive ST-80-MRSA-IV followed by PVL-positive ST-30-MSSA (24). In Algeria, it was reported that ST-80-MRSA-IV was present in most neonates tested over an 18-month period, with a minority being PVL-positive (31). Finally, for Egypt, it was reported that the prevalent MLSTs are ST-30, ST-80, and a novel type, ST-1010; PVL prevalence was estimated at 19% (32). Enany et al. reported that the Egyptian ST-80 lineage was different from the globally prevalent ST-80, primarily due to a unique *spa* type and antimicrobial resistance (32).

While extensive surveys have provided insight into the prevalence and genotypes of MRSA in Europe (33), Asia (34, 35), and North America (36, 37), limited data are

available for Egypt and the rest of the Middle East (23). Because of its central location, as well as its political and historical role, Egypt presents a unique case study for staphylococcal distribution and exchange in the Middle East (38). Egypt's cultural and geographical placement may facilitate local staphylococcal exposure to international lineages from other parts of the Middle East, as well as Asia, Europe, and Africa. The accessibility of WGS presents an opportunity to profile staphylococci in Egypt and more broadly the Middle East in terms of gene marker typing, core genome, and phylogenomics. Prior to this study, there were only 34 *S. aureus* and 6 *S. haemolyticus* genome assemblies for isolates from the Middle East. Here, we report the phylogenetic and phylogenomic associations of 56 *S. aureus* and 10 *S. haemolyticus* isolates from Egypt, describing the first comparative genomic study of these two species, including the 66 newly sequenced strains and the 40 previously deposited assemblies. WGS afforded insight into the lineage and genetic content of these two staphylococcal species, including type information historically obtained using molecular methods.

## RESULTS

**Strain genotyping.** We produced draft genomes of 56 *S. aureus* and 10 *S. haemolyticus* isolates from the Alexandria Main University Hospital (AMUH) in Egypt. Genome assembly statistics and metadata are available in Table S1. Complementing these new genome assemblies, our analyses included publicly available genome assemblies from strains isolated in the Middle East, including 34 *S. aureus* strains (from Egypt, $n = 17$, Kuwait, $n = 5$, Lebanon, $n = 4$, Tunisia, Palestine, and United Arab Emirates, $n = 2$ each, and Morocco and Sudan, $n = 1$ each) and 6 *S. haemolyticus* strains (all from Egypt).

The genomes represent varied MLSTs. The 16 *S. haemolyticus* isolates examined here belonged to 9 MLSTs, including a new sequence type ST-74 (strain 51) assigned as a result of this study and an isolate of unknown sequence type (ST) (strain 7A). ST-3 was the most common among the isolates examined ($n = 4$) (Table S2). A total of 20 *S. aureus* MLSTs were identified, including 3 novel types: ST-5860 (strain 48), ST-5861 (strain 2705404), and ST-5862 (strain 2705410); all 3 of these strains came from prior studies and were isolated from Egypt, Kuwait, and Lebanon (Table S2). Twelve different MLSTs were identified among the Egyptian isolates; ST-239 was the most frequently identified type ($n = 24$), followed by ST-1 ($n = 19$) and ST-80 ($n = 12$). Two of the AMUH *S. aureus* isolates, strains AA32 and AA35, could not be typed due to incomplete sequences. For AA32, the closest MLST match was ST-22, showing partial sequences for 2 of the 7 housekeeping genes constituting the MLST typing scheme of *S. aureus*.

*S. aureus* isolates are routinely described by their clonal complex (CC), each comprising several different sequence types. The *S. aureus* genomes examined here could be categorized into seven CCs (Table S2), the largest being CC8 ($n = 26$), consisting mainly of Egyptian isolates and one Moroccan isolate (strain 12480433). Twelve strains were identified as ST-80, a sequence type that does not belong to any of the defined CCs. Twenty different *spa* types were identified in addition to 7 isolates that could not be typed. The *spa* type t037 was most abundant among the strains examined here ($n = 23$), with all but one belonging to CC8; furthermore, 20 of these 23 isolates belonged to ST239-SCC*mec* III clone. The next most frequent *spa* type identified was t127 ($n = 19$), all belonging to CC1. Table 1 summarizes these results.

**Core and pangenomes of *S. haemolyticus* and *S. aureus* strains.** To investigate the core genome and pangenome of *S. haemolyticus*, we included the 6 publicly available *S. haemolyticus* genomes from Egypt to the 10 newly sequenced *S. haemolyticus* genomes (Table S1). The pangenome for these strains included 3,541 genes present in one or more of the genome assemblies (Fig. S1), with 1,834 single-copy-number genes in the core genome. Included within these core genes are the virulence factors autolysin (*atl*), elastin binding protein (*ebp*), thermonuclease (*nuc*), and cytolysin (*cylR2*).

Our analysis of the core genome and pangenome of *S. aureus* included both our 56 *S. aureus* genomes and the 34 publicly available *S. aureus* draft genome assemblies from the Arab region (Table S1). The pangenome of these 90 strains contained 4,283 genes (Fig. 1A), and the core genome included 1,501 single-copy-number genes. The

**TABLE 1** MLST clonal complexes, *spa* types, and SCC*mec* types among the *S. aureus* isolates

| MLST CC | Strain name | Geographical origin | *spa* type[a] | SCC*mec* type[a] |
|---|---|---|---|---|
| CC1 | 3 (a), 3 (B), 23, 6 (B), 43, AA51, AA67, AA77 | Egypt | t127 | N/D |
| | R181, R180, AA1, AA78 | UAE, Egypt | t127 | N/D |
| | 6 (a), AA69 | Egypt | t127 | N/D |
| | AA59, AA65, AA68 | Egypt | t127 | No *mecA* detected |
| | AA103, AA87 | Egypt | t127 | No *mecA* detected |
| CC15 | 15, 16 | Egypt | t094 | No *mecA* detected |
| | 17 | Egypt | Unk | No *mecA* detected |
| CC22 | 41 | Egypt | t13828 | IV |
| | Gaza_MRSA_B62 | Palestine | t223 | IV |
| | AA18 | Egypt | t223 | IV |
| | AA5 | Egypt | t3243 | IV |
| | Gaza_MRSA_B04 | Palestine | t790 | IV |
| | 40 | Egypt | Unk | IV |
| CC30 | AA41 | Egypt | t037 | No *mecA* detected |
| | 19 | Egypt | t1504 | N/D |
| CC5 | AA30 | Egypt | t304 | IV |
| | 14, AA76, AA80 | Egypt | t688 | N/D |
| | AA70 | Egypt | t688 | VI(4B) |
| CC8 | 12480433 | Morocco | t008 | IV |
| | LHI_Sa_30 | Egypt | t008 | N/D |
| | 46 | Egypt | t030 | III |
| | 50, AA101, AA13, AA14, AA22, AA23, AA27, AA31, AA33, AA46, AA52, AA55, AA57, AA60, AA61, AA62, AA63, AA64, AA91, AA92 | Egypt | t037 | III |
| | AA79 | Egypt | t037 | N/D |
| | AA93 | Egypt | t037 | No *mecA* detected |
| | AA29 | Egypt | Unk | III |
| CC97 | AA36 | Egypt | t267 | N/D |
| | AA39, AA6 | Egypt | t267 | IV |
| | AA104 | Egypt | t267 | N/D |
| | AA8 | Egypt | Unk | IV |
| ST-80 | 2705432, 2705403, 2705405, 2705407, 2705409, 2705412 | Tunisia, Kuwait, Lebanon | t044 | IV |
| | AA45 | Egypt | t044 | IV |
| | 2705431 | Tunisia | t044 | No *mecA* detected |
| | 2705411 | Lebanon | t131 | IV |
| | AA2 | Egypt | t416 | IV |
| | AA3, AA4 | Egypt | t416 | N/D |

[a]Unk, unknown; N/D, not determined.

functionality of the genes within the *S. aureus* core genome was further investigated, based upon their COG categories (Fig. 1B). The core genome was further examined for virulence factors, and we found that the gene related to autolysin (*atl*) in *S. haemolyticus* was also found in *S. aureus*, as well as genes associated with intercellular adhesin, cysteine protease, thermonuclease, capsule, and the type VII secretion system (Table 2). The accessory genome contained 2,178 genes; these genes were present in one or more of the genomes but not all of them. This large accessory genome suggests that the Middle Eastern isolates have an open pangenome.

We also identified virulence factors and antibiotic resistance genes within the accessory genome (Tables S3 and S4). Thirty-one of the *S. aureus* genomes were positive for *lukF/S*-PV, which encodes PVL; this includes 12 of the newly sequenced strains. Isolates positive for PVL were mainly (77%) *mecA* positive, present in CC1, ST-80, CC30, and CC8, and obtained from Egypt, Kuwait, Tunisia, Lebanon, and Morocco. Importantly, isolates obtained from CA infections belonged to CC1, ST-80, CC5, CC97, and CC8, making PVL presence a good predictor for the ability of the isolate to cause CA infections.

**Phylogenomic study of *S. haemolyticus* and *S. aureus* strains from the Middle East.** The core genes were used to derive phylogenies for each species. The *S. haemolyticus* isolates were all from Egypt and clustered into two clades (Fig. 2). As the tree shows, variation between the core genomes was minor. Furthermore, the clade structure

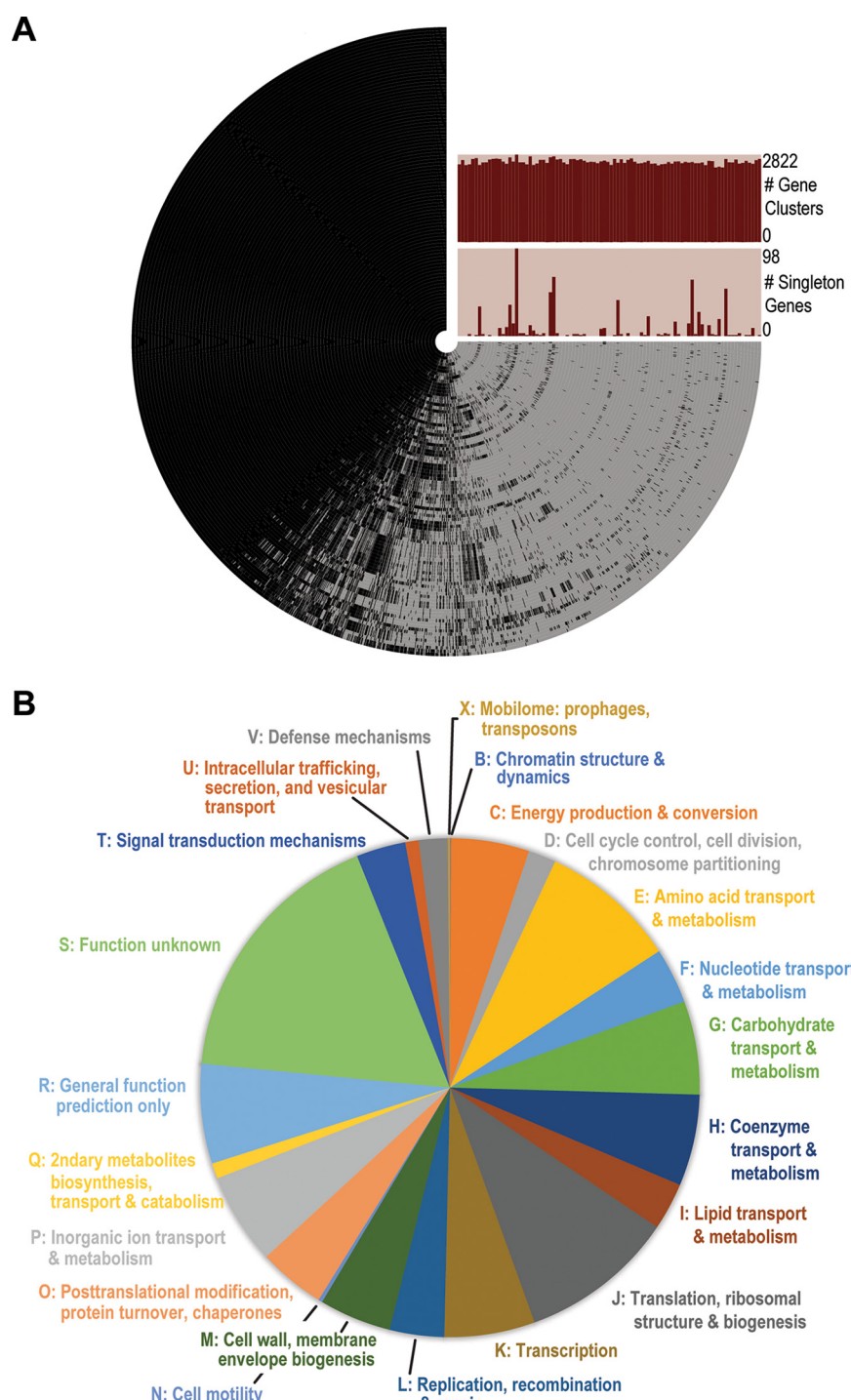

**FIG 1** Genome analysis of 90 *S. aureus* isolates recovered from the Middle East. (A) The pangenome. Each ring corresponds to a single genome. Each radial extension in the ring corresponds to the presence (black) or absence (light gray) of a given gene cluster (homologous gene). The bar charts list the number of genes identified in the given genome (top) and the number of singleton genes or genes that are unique to the given genome (bottom). The pangenome of these 90 isolates contained 4,283 genes, the core genome included 1,501 single-copy-number genes, and the accessory genome contained 2,178 genes. (B) Functionality of genes contained within the core genome. The same autolysin gene (*atl*) found in the core genome of *S. haemolyticus* was found in *S. aureus*.

**TABLE 2** Virulence factors included in the *S. aureus* core genome

| VFclass | Virulence factor | Related gene(s) |
|---|---|---|
| Adherence | Autolysin | *atl* |
| | Intercellular adhesin | *icaA*, *icaD*, *icaR* |
| Enzyme | Cysteine protease | *sspC* |
| | Thermonuclease | *nuc* |
| Immune evasion | Capsule | *cap5A*, *cap8B*, *cap5M*, *cap8N*, *capO* |
| Secretion system | Type VII secretion system | *esaB*, *essA*, *essB*, *esxA* |

of the genomes corresponded with MLST, indicated in the bar of Fig. 2. The MLST tree for these genomes is shown in Fig. S2.

*S. aureus* isolates came from all over the Middle East and clustered into 6 clades, indicated by the grayscale bar in Fig. 3, with Egyptian isolates represented in all clades (Fig. 3 and 4). Clade 1 isolates belonged to ST-1 and were from Egypt and the United Arab Emirates, and clade 2 contained most of the publicly available strains, including 4 strains sequenced by our group. The predominating subclone seen among 46.7% of the isolates within clade 2 was *spa* t044/SCC*mec* IV/ST-80, which shows some degree of shared content between these isolates. Clade 3 isolates were solely from Egypt and belonged mainly to ST-15 and ST-5. Clade 4 comprised isolates from Egypt, Sudan, and Palestine, with most belonging to ST-22 and ST-361. Clade 5 contained isolates from Egypt, belonging mainly to ST-97. The remaining isolates were in clade 6, of ST-239, and from Egypt, except for one Moroccan isolate. This clade represents a *spa* t037/ST239-SCC*mec* III/MLST CC8 clone. The phylogenetic tree derived from the core genome sequences corresponded with the tree derived from the MLST marker genes (Fig. S2). Fourteen isolates lacked *mecA* (Fig. 4, pale green star) and occurred predominantly in CC1 (*n* = 5), CC15 (*n* = 3), and CC30, CC8, and ST-80, with one isolate in each; in addition, three isolates belonged to ST-361 (*n* = 2) or ST-5860 (*n* = 1). Thirteen of these *mecA*-negative isolates were from Egypt.

Next, we compared the Middle Eastern *S. aureus* and *S. haemolyticus* isolate genomes to genomes of isolates collected from Europe and Asia. We restricted our analysis to isolates collected from 2010 through 2019, as the Egyptian isolates sequenced in this study were collected in 2015. In total, 302 *S. aureus* and 82 *S. haemolyticus* genomes were included in this analysis (see Materials and Methods). The core genome was computed for both species, identifying a single-copy core genome of 445 genes and 1,071 genes for *S. aureus* and *S. haemolyticus*, respectively. Based upon these core genomes, the phylogenies were derived, indicating the continent of origin for each genome (Fig. 5).

## DISCUSSION

Prior to the study initiated here, there were limited genomic data for *S. aureus* and *S. haemolyticus* isolates from the Middle East. The addition of the 56 *S. aureus* and 10 *S. haemolyticus* genomes produced here significantly increased the number of available genomic sequences from this region and enabled this first investigation of strain diversity within the region, including genomes of strains isolated from all over the Middle East.

We found that several different MLSTs and CCs are circulating within Egypt and more broadly within the Middle East. The CCs we identified were quite diverse, with the isolates grouped into 6 clades, which were more closely related within the broader context of the CC. We identified 20 *S. aureus* MLSTs in the region, including 3 new sequence types. Twelve of these MLSTs include isolates from Egypt. Our isolates, as well as others from the Middle East, were assigned to ST-30, which has spread through Asia (39), ST-22, which is the dominant HA-MRSA type in Europe (39), and ST-80, the dominant CA-MRSA type in Western Europe (39). Four of the newly sequenced strains, as well as publicly available strains from Tunisia, Kuwait, and Lebanon, are members of ST-80. The identification of isolates from Egypt that are members of the predominant

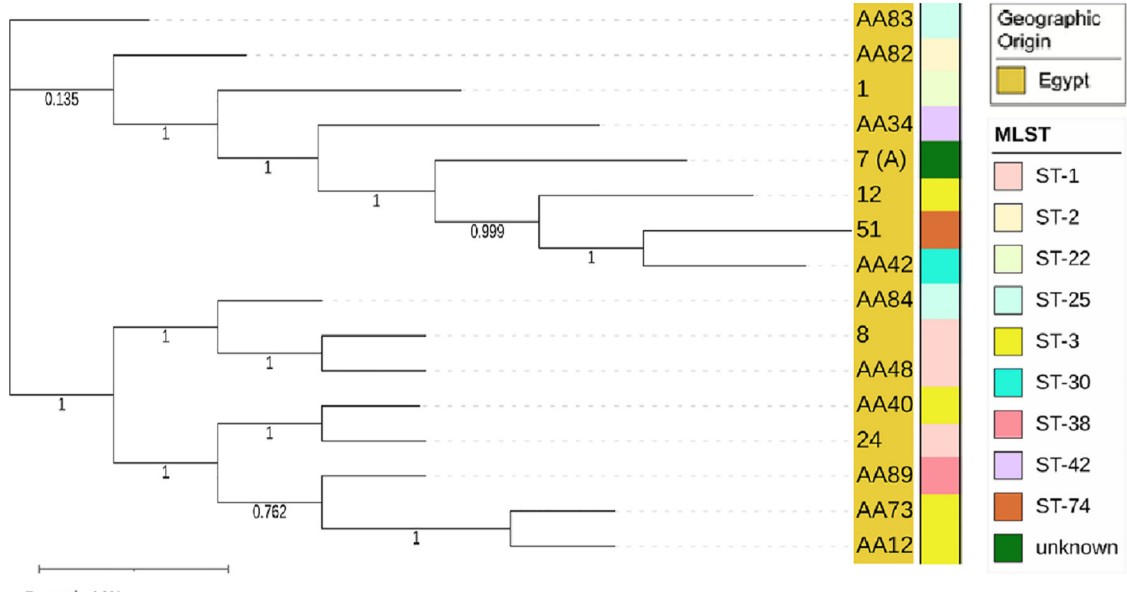

**FIG 2** Phylogeny based upon the core genes for the *S. haemolyticus* isolates. All *S. haemolyticus* isolates were from Egypt and clustered into two clades corresponding with MLST.

strains in Europe and Asia suggests that Egypt's geographical location plays an important role in the distribution and exchange of lineages between these two continents. Analysis of the *S. haemolyticus* genomes found 9 MLSTs in circulation within Egypt, including one new sequence types. Because few isolates are publicly available for *S. haemolyticus* through PuBMLST (*n* = 166; as of March 2022) and no continental surveys have been conducted to date, it is not possible to ascertain if Egyptian strains include predominant strains in circulation in Asia and Europe.

The core genome for the *S. aureus* strains examined here is slightly larger than that previously calculated for the species (40, 41). This is expected, as our analysis is restricted to fewer genomes from a single region. Both our *S. haemolyticus* and *S. aureus* core genomes include *atl*, a gene that is essential for biofilm formation (42). The *S. aureus* core genome also includes the *ica* gene cluster, which is also associated with biofilm formation (43, 44), as well as its regulator *icaR* (45). The presence of *atl* and the *ica* gene cluster signifies the biofilm potential of the isolates.

Our analysis of the Middle Eastern *S. aureus* genomes finds an open pangenome, which concurs with prior comparative genomic studies for this species (40). Horizontal gene transfer (HGT) between strains, coagulase-negative *Staphylococcus* (CoNS) strains, and other species is well documented (see review [46]). Virulence factors and antibiotic resistance genes are prime candidates for HGT and were found within the accessory genome of the Middle Eastern strains (Tables S3 and S4). Some of these virulence factors are carried by prophages, e.g., PVL, or plasmids, e.g., *blaZ* (47, 48). There is growing concern about PVL in Egypt; a recent study found that PVL-positive MRSA strains are prevalent in retail unpasteurized cow's milk (49). However, a retrospective study of *S. aureus* isolates from 250 septic pediatric patients in one Egyptian hospital found that only 4% were PVL-positive (50).

Phylogenomic analyses of the core genome largely mirrored MLST types regardless of geographic origin (Fig. 3). Strains of the same SCC*mec* type had a more similar core genome sequence (Fig. 4). In a prior phylogenetic study, John and coworkers found that 16S rRNA gene sequence similarity did not correspond with SCC*mec* type, leading them to conclude that horizontal gene transfer plays a role in resistance gene acquisition (40). Recently, Soliman et al. published a study characterizing the genomes of 18 MRSA isolates from a tertiary care hospital in Cairo, Egypt; their isolates were primarily

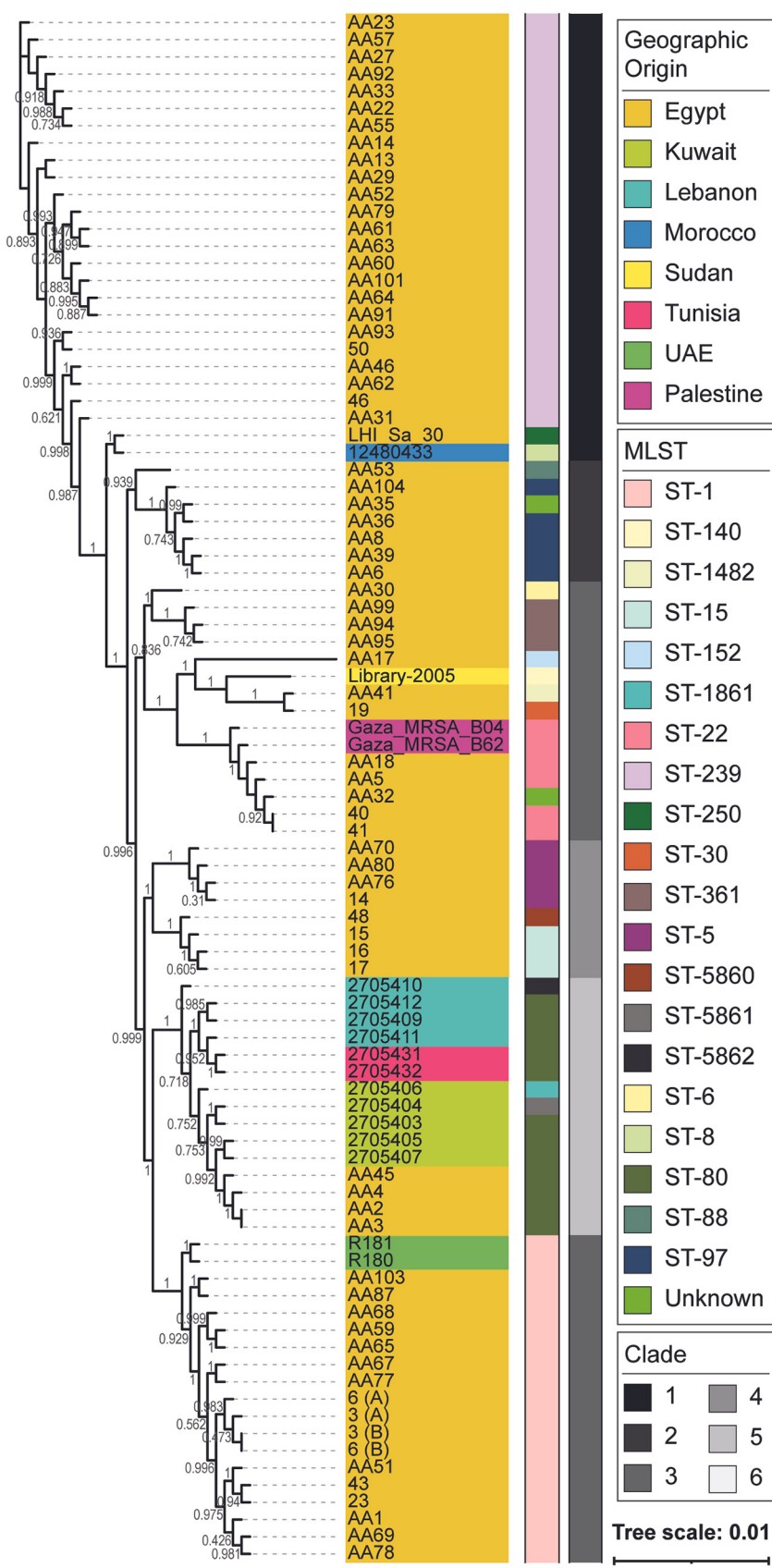

**FIG 3** *S. aureus* core genome phylogeny colored by geographical origin of isolation (strain name color) and MLST (middle bar) and clade (right bar). *S. aureus* isolates were from different parts of the

SCC*mec* types V (*n* = 9) and VI (*n* = 2), not observed within our larger collection (51). In contrast, our study found that SCC*mec* types III and IV were equally prevalent within the region. Similarly, SCC*mec* types V and IV have been observed most frequently in other *S. aureus* studies within the region (52–54). SCC*mec* type III predominated among HA-MRSA infections, suggesting that, in contrast to these prior studies, our isolates indicate that HA infections have a higher incidence among the patients tested here. AMUH, where our isolates were collected, is the largest tertiary hospital and main referral center in the Northern sector of Egypt; thus, patients with more severe infections would be more likely to be treated at AMUH than at any other hospital in Northern Egypt. Prior studies have found SCC*mec* type III to be the predominant type in Asian countries (34). The fact that the SCC*mec* type III/MLST ST-239 is the oldest pandemic strain of MRSA (55) might explain its prevalence among the current collection of isolates. The *spa* t044/SCC*mec* IV/ST-80 clone seen among 6 Middle Eastern isolates and 1 Egyptian isolate in clade 2 matches the European clone (56), with another isolate from Tunisia lacking *mecA* and thus potentially constituting a modified *S. aureus* (MODSA) strain. The remaining SCC*mec* IV/ST-80 isolates were of a different *spa* type for all Egyptian isolates sequenced in the current study. Furthermore, they carried fusidic acid and tetracycline resistance genes, consistent with the European clone and different from the clone described by Enany et al. (32), which was of *spa* type t042 and had a different resistance profile.

Exploring the genomic diversity of strains in Egypt and the Middle East is a critical first step for future studies considering the spread and evolution of staphylococci isolated from the region between Asia and Europe. As our core genome comparison of isolates from Asia, Europe, and the Middle East shows, most of the *S. aureus* clades include genomes from all three regions (Fig. 5A). This suggests multiple instances in which *S. aureus* spread throughout the regions. We can also observe clades containing only isolates from Asia and Europe. Similarly, we observe clades in the *S. haemolyticus* tree (Fig. 5B) that contain only genomes of isolates from Asia and Europe. While it may be that these strains spread directly between Asian and European countries, it is important to keep in mind that in contrast to many European and Asian countries, there are few if any isolates sequenced from the Middle East. To more definitively ascertain the role that Middle Eastern countries, including Egypt, play in the transmission of staphylococci between Asia and Europe, additional isolation and sequencing are needed.

The *S. haemolyticus* and *S. aureus* genomes examined here provide insight into the diversity of strains currently in circulation in Egypt, particularly with respect to their carried virulence factors and antibiotic resistance genes. WGS analysis enabled a more complete picture of this diversity than molecular typing strategies. Our analysis of the *S. haemolyticus* genomes is the first of strains isolated in Egypt. Future studies to catalog the diversity of *S. aureus* and *S. haemolyticus* strains circulating in the Middle East are desperately needed. Identifying the main genotypes, as well as the resistance and virulence mechanisms among the resistant isolates in the region, can drive antibiotic stewardship and infection control plans. The Middle East, especially Egypt, is likely an important geographical location for dissemination of endemic staphylococci. Future surveys of strains in circulation in the Middle East will help determine if strains are transmitted to and/or from Asia and Europe through the Middle East.

## MATERIALS AND METHODS

**Bacterial isolates.** Fifty-six *S. aureus* and 10 *S. haemolyticus* consecutive nonduplicate isolates were collected from the Medical Microbiology Laboratory at AMUH between September and December 2015.

**FIG 3** Legend (Continued)
region and clustered into six clades, each containing Egyptian isolates. Clade 1 isolates belonged to ST-1 and were from Egypt and the United Arab Emirates, and clade 2 contained the majority of the Arab isolates, with *spa* t044/SCC*mec* IV/ST-80 as the predominating clone. Clade 3 isolates were solely from Egypt and belonged mainly to ST-15 and ST-5. Clade 4 comprised isolates from Egypt, Sudan, and Palestine, with the majority belonging to ST-22 and ST-361. Clade 5 contained isolates from Egypt and belonged mainly to ST-97. The remaining isolates were in clade 6, of ST-239, and from Egypt and Morocco (*n* = 1). This clade represents a *spa* t037/SCC*mec* III/MLST CC8 clone.

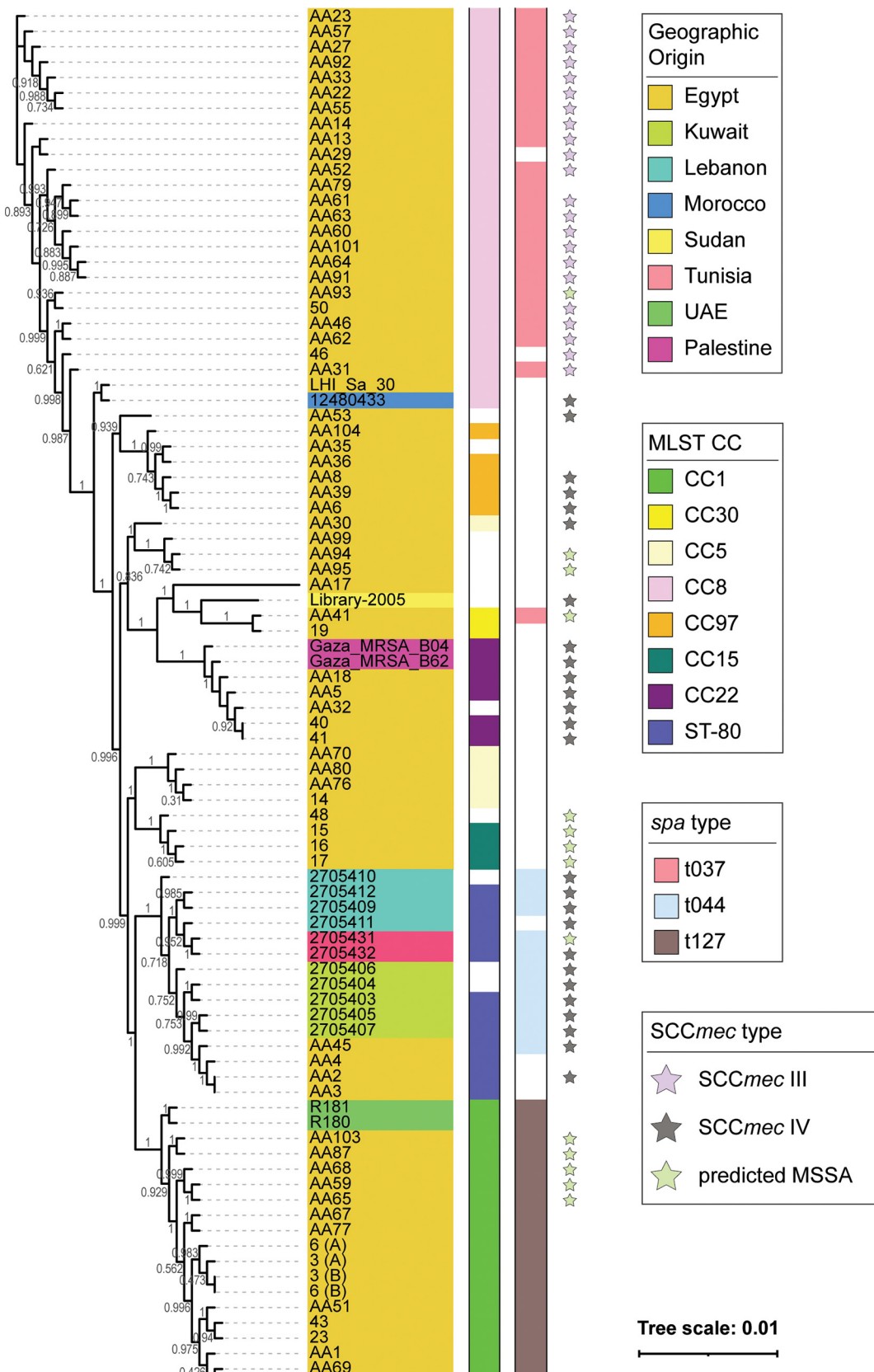

**FIG 4** Phylogenetic tree based on core genome annotated by geographic origin, MLST CC, and main *spa* and SCC*mec* types. Fourteen isolates, mostly from Egypt, lacked *mecA* and occurred predominantly in CC1 (*n* = 5), CC15 (*n* = 3), CC30, CC8, and ST-80 (one isolate in each), ST-361 (*n* = 2), or ST-5860 (*n* = 1).

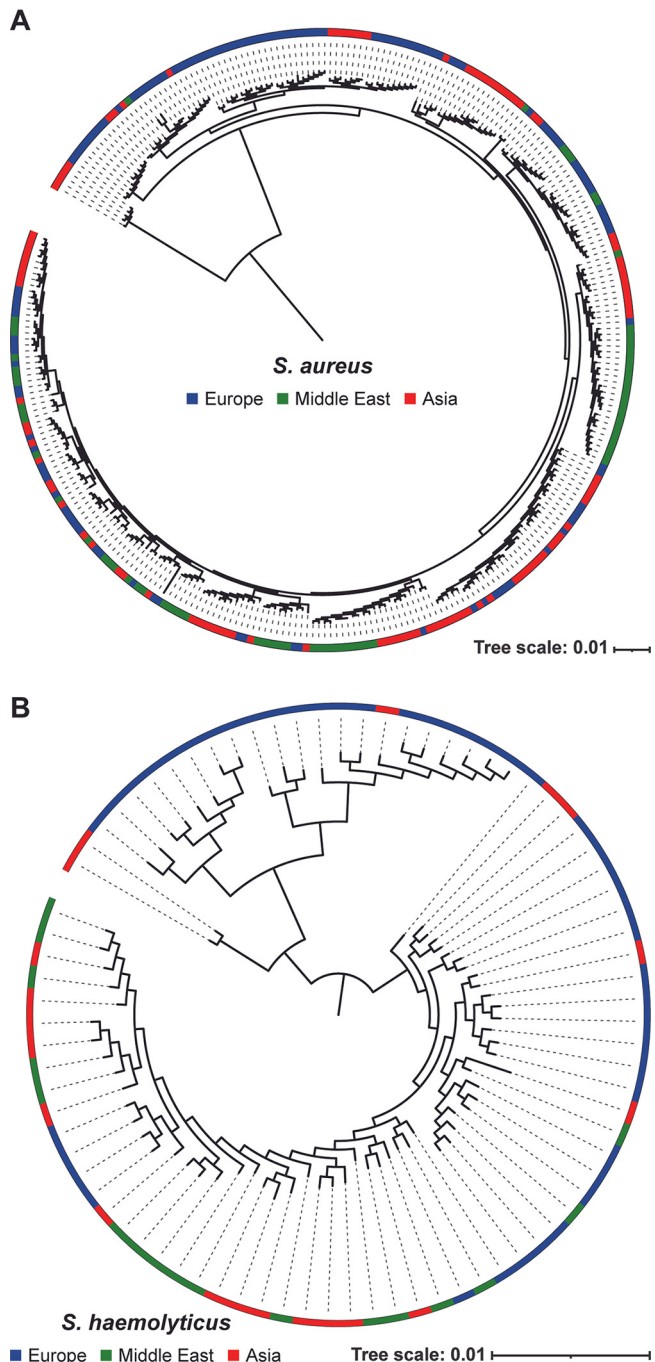

**FIG 5** Phylogenetic trees based on core genome annotated by continent of origin for (A) *S. aureus* and (B) *S. haemolyticus* isolates. The continent of isolation for each genome included in these analyses is indicated by the color in the outer band of the circular tree with Europe in blue, Middle East in green, and Asia in red.

These isolates were obtained from various clinical specimens, including pus, blood, sputum, urine, tissue, aspirate, and broncho-alveolar lavage (BAL) fluid. All isolates were identified using conventional methods, such as Gram staining, growth on and fermentation of mannitol salt agar (MSA; Oxoid Ltd., England), growth on DNase agar, and slide coagulase testing using Dryspot Staphytect Plus (Oxoid Ltd., England), and confirmed using matrix-assisted laser desorption ionization–time of flight mass spectrometry (MALDI-TOF MS) with the MALDI Biotyper 3.0 software (Bruker Daltonik, USA). The isolates were further classified as either hospital-acquired or community-acquired infections based on a 48-h window between the dates of patient admission and isolate collection (57).

**DNA extraction.** Colonies grown on tryptone soy agar (TSA) plates were harvested and washed in 1 mL phosphate-buffered saline (PBS) and resuspended in 0.5 mL SET (75 mM NaCl, 25 mM EDTA, 20 mM Tris [pH 7.5]), to which 50 $\mu$L of fresh 20 mg/mL lysozyme in PBS and 30 $\mu$L Mutanolysin were added; the mixture was incubated at 37°C for 60 min. The cells were then treated with 60 $\mu$L 10% sodium dodecyl sulfate and 20 $\mu$L proteinase K and incubated at 55°C for 2 h with gentle inversion. The suspension was mixed gently with 210 $\mu$L of 6 M NaCl, and 700 $\mu$L phenol:chloroform was added, followed by incubation at room temperature for 30 to 60 min, using a rotating wheel for gentle mixing. The suspension was then centrifuged at maximum speed for 10 min, and the aqueous phase was transferred to a new microcentrifuge tube and mixed gently with an equal volume of isopropanol. The tubes were centrifuged to produce a DNA pellet that was washed with 70% ethanol, which was left to evaporate overnight. The pellets were resuspended in 50 $\mu$L double-distilled water (ddH$_2$O) and stored at −20°C until further processing.

**Genome sequencing and genome assembly.** The Illumina Nextera kit was used for whole-genome library preparation. Each isolate was sequenced using the Illumina MiSeq System, producing paired-end 2 × 250 bp reads. Quality control and demultiplexing of sequence data were done with onboard MiSeq Control software and MiSeq Reporter v3.1. Raw reads were trimmed using Sickle v1.33 (https://github.com/najoshi/sickle), and the trimmed reads were assembled using SPAdes v3.13.0 (58) with the "only-assembler" option for *k* values of 55, 77, 99, and 127. Genome coverage was calculated using BBMap v38.47 (https://sourceforge.net/projects/bbmap/). Contigs shorter than 500 bp were removed from the assemblies using bioawk (https://github.com/lh3/bioawk). Genome assemblies were annotated, species was confirmed, and the quality of the assembly was determined using PATRIC v3.3.18 (59). As part of the annotation process, PATRIC performs checkM (60), and we used these data to confirm species designation and genome completeness and contamination. Genome assemblies were deposited in NCBI's Assembly database, along with raw sequence data in SRA under BioProject accession number PRJNA648411. Deposited genomes were annotated using the NCBI Prokaryotic Genome Annotation Pipeline (PGAP) v5.0 (61). Unless previously noted, default parameters were used for each software tool.

To complement our analysis of the genomes from AMUH, raw sequence data for 41 *S. aureus* and 10 *S. haemolyticus* strains were retrieved from NCBI. These records were identified by searching SRA (as of January 2020) for strains isolated in the Arab region. These raw reads were processed as indicated above. High-quality assemblies were included in subsequent analyses.

**Bioinformatic analysis.** Multilocus sequence typing (MLST) was determined using the MLST v2.0.4 web server available through the Center for Genomic Epidemiology (62). This web-based tool utilizes the MLST allele sequence and profile data from PubMLST (v2.0.0) (63). For *S. aureus* assemblies, the *S. aureus* MLST configuration was used, and for the *S. haemolyticus* assemblies, the *S. haemolyticus* configuration was used. *spa* typing was performed using the online tool SpaTyper v1.0 available through the Center for Genomic Epidemiology (64). SCC*mec* typing was performed using SCCmecFinder v1.2 online tool available through the Center for Genomic Epidemiology (https://cge.food.dtu.dk/services/SCCmecFinder/) (65, 66); default parameters (90% threshold for percent identity [%ID] and 60% minimum length) were used. Resistance and virulence genes were identified using PATRIC v3.6.5 (67) and VFAnalyzer (68). For VFAnalyzer analysis, the *Staphylococcus* genus was specified.

**Phylogenomic and phylogenetic analysis.** The core and pangenomes were generated using anvi'o v5.1. The following anvi'o functions were used to calculate the pangenome: anvi-gen-genomes-storage, anvi-pan-genome with the parameter –mcl-inflation 8, and anvi-display-pan (69, 70). To obtain the single-copy core genome, the anvi-get-sequences-for-gene-cluster function was used with the parameters –min-num-genomes-gene-cluster-occurs N –max-num-genes-from-each-genome 1, where "N" is the number of genomes for *S. aureus* or *S. haemolyticus*. Functional groups for the core genome were determined by querying core genome amino acid sequences against the COG database (71) through anvi'o. The core genes were concatenated for each genome and then aligned using MAFFT v7.388 (72) through anvi'o with default parameters. The tree was built using the FastTree v2 (73) plugin in Geneious Prime v2019.2 (Biomatters Ltd., Auckland, New Zealand) with default parameters. MLST sequences were downloaded from PubMLST v2.0.0 (63) and aligned in Geneious Prime v2019.2, and the trees were built using the FastTree v2 (73) plugin in Geneious Prime v2019.2, again with default parameters. iTOL v5.6.1 (74) was used to annotate and visualize all trees.

Intercontinental comparisons of *S. aureus* and *S. haemolyticus* genomes were also conducted. All publicly available genomes for these two species were retrieved from NCBI. Based upon the metadata associated with each genome, we restricted our analysis to strains collected between 2010 and 2019 and from a country/region in Asia, Europe, or the Middle East. The collection date range was implemented given that the Egypt isolates were collected in 2015. All publicly available *S. haemolyticus* genomes meeting these criteria were included in the analysis as well as the six previously deposited *S. haemolyticus* genomes from Egypt for which no collection date was available. This data set thus includes 82 genomes. In total, 1,790 *S. aureus* genomes met the date and country/region criteria. These genomes were subsampled such that, for isolates from Asia and Europe, three genomes were randomly selected for each year and country/region combination; all isolates from the Middle East were included. In total, 302 genomes were selected for analysis. Table S5 lists details about the genomes included for both species. For the intercontinental *S. aureus* and intercontinental *S. haemolyticus* genomes, the core genome was identified via anvi'o, and a phylogenetic tree was derived via FastTree and visualized via iTOL as described above.

**Data availability.** Raw sequencing reads and assembled genomes can be found at BioProject accession number PRJNA648411.

## SUPPLEMENTAL MATERIAL

Supplemental material is available online only.

**SUPPLEMENTAL FILE 1**, PDF file, 0.4 MB.
**SUPPLEMENTAL FILE 2**, XLSX file, 0.1 MB.
**SUPPLEMENTAL FILE 3**, XLSX file, 0.02 MB.

## ACKNOWLEDGMENTS

We thank Thomas Halverson for the DNA extraction, Roberto Limeira and Loyola Genomics Facility for performing the whole-genome sequencing of the isolates, and Adriana Ene for her assistance with the intercontinental analyses. We also acknowledge funding from NIH (R01 DK104718 awarded to A.J.W.), NSF (1661357 awarded to C.P.), USAID (GSP-T85 awarded to A.A.), and DFG (ZI 665/3-1 awarded to A.A.). The funders did not play a part in the design or conduct of the study.

C.M.: formal analysis, writing—original draft preparation, writing—review and editing. C.R.M.: formal analysis, writing—original draft preparation, writing—review and editing. C.P.: formal analysis, writing—original draft preparation, visualization, writing—review and editing. A.J.W.: formal analysis, conceptualization, writing—review and editing. A.A.: conceptualization, formal analysis, writing—original draft preparation, writing—review and editing.

A.J.W. is a member of the Advisory Board of Urobiome Therapeutics. The remaining authors report no competing interests.

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
