## [Reviewer comments · Microbiology Spectrum]

Microbiology Spectrum

Whole genome sequencing of *Staphylococcus aureus* and *Staphylococcus haemolyticus* clinical isolates from Egypt

Cesar Montelongo, Carine Mores, Catherine Putonti, Alan Wolfe, and Alaa Abouelfetouh

Corresponding Author(s): Alaa Abouelfetouh, Faculty of Pharmacy, Alexandria University and Alamein International University

Review Timeline:

Submission Date:	November 28, 2021
Editorial Decision:	January 4, 2022
Revision Received:	April 17, 2022
Editorial Decision:	April 24, 2022
Revision Received:	May 25, 2022
Accepted:	May 31, 2022

Editor: *Hermine Mkrtchyan*

Reviewer(s): *The reviewers have opted to remain anonymous.*

Transaction Report:

DOI: <https://doi.org/10.1128/spectrum.02413-21>

Dr. Alaa Abouelfetouh
Faculty of Pharmacy, Alexandria University and AlAlamein International University
Microbiology and Immunology
1 Khartoum Square, Azarita
Alexandria 21521
Egypt

Re: Spectrum02413-21 (Phylogenomic study of *Staphylococcus aureus* and *Staphylococcus haemolyticus* clinical isolates from Egypt)

Dear Dr. Alaa Abouelfetouh:

I have received the reviews of your manuscript entitled "Phylogenomic study of *Staphylococcus aureus* and *Staphylococcus haemolyticus* clinical isolates from Egypt", and I regret to inform you that we will not be able to publish it in Spectrum. Your submission was read by reviewers with expertise in the area addressed in your study and it was the consensus view of these reviewers that your paper did not meet the standards necessary for publication. Copies of the reviewers' comments are attached for your consideration.

I am sorry to convey a negative decision on this occasion, but I hope that the enclosed reviews are useful. Please note, rejections from Microbiology Spectrum are final and your manuscript will not be considered by other ASM journals. We wish you well in publishing this report in another journal and hope that you will consider Spectrum in the future.

Sincerely,

Hermine Mkrtchyan
Editor, Microbiology Spectrum

Reviewer comments:

Reviewer #1 (Comments for the Author):

General comment

In their study, authors aimed the use of genomic data to investigate circulating strains of *S. aureus* and *S. haemolyticus* in Egypt. Despite the strength of genomic data provided, the scientific content, the structure as well as the wording of this article need to be significantly improved. English revision should be done for resubmission. Example: Line 74 (the infection it can cause range)

Specific comments

The text of the Impact statement remains very descriptive and does not raise the impact of this study.

The Introduction should end at line 130. The following text describes the results.

Line 124: Middle East and elsewhere. What does elsewhere mean? Africa and Mediterranean?

The results section begins with a paragraph without a subtitle which ultimately describes more of a methodology and very few of the results.

The discussion, which should be a discussion of the results, relates more elements that have their place in the introduction. The discussion is not hard-hitting and should be better developed.

In Methods (Bacterial isolates), the authors speak of 89 *S. aureus* and 14 *S. Haemolyticus*, which does not correspond to the figures described in the abstract.

In this same paragraph, the authors describe poorly performing conventional methods for the identification of the genus *Staphylococcus* identification. I would have expected confirmation based on the genomic data in their possession.

The low number of non-Egyptian genomes in this study cannot allow a prevalence comparison among MLST types. Authors should use this genomic data with another angle.

Reviewer #2 (Comments for the Author):

The research article proposed by Montelongo et al entitled "Phylogenomic study of *Staphylococcus aureus* and *Staphylococcus haemolyticus* clinical isolates from Egypt" describes one of the important topics.

In general the article is of great impact to readers working in the healthcare settings, especially as considering it performing the WGS for large number of isolates, however, some minor issues need to be addressed first before considering final publication. Those are shown below.

1- The author need to add in the introduction section a short paragraph about the clinical outcomes and the PVL positive *S. aureus* specially in Egypt.

2- PVL positive isolates are mainly accompanied with severe outcomes like the brain abscess so illustrating such correlation will raise the significant of the work.

3- The authors need to emphasize that the geographical location of Egypt plays an important role in the dissemination of various lineage and types within this country.

4- Please rewrite the part of the open pangenome to be simpler and clearer to the readers.

5- Table 2: Are these all the identified list of the virulence factors?? too little.

Reviewer #3 (Comments for the Author):

In this manuscript, *Staphylococcus aureus* and *Staphylococcus haemolyticus* clinical isolates from Egypt were collected. In total, 56 *S. aureus* and 10 *S. haemolyticus* isolates were collected and also performed WGS. *Staphylococci* are an important heterogeneous group of pathogens which can cause severe infection. The study of antibiotic resistance and virulence for these two strains can bring a large impact for the clinical research. However, only minority of *S. aureus* and *S. haemolyticus* studies provided the information from Middle East which is the connection between Asia and Europe. The authors of this manuscript cooperated Alexandria Main University Hospital to fill in this gap. All the WGS data including contigs, scaffolds and annotations are opened for public usages. Moreover, the antibiotic-associated and virulence-associated genes also provide useful information for the future clinical study. Furthermore, the phylogenomic study revealed the relationship between region and MLST as well.

Based on the manuscript, I have several suggestion and questions:

1. For the reproducibility, the parameters that the authors assigned for the software should be provided. If the default setting was used, it also needs to be written in the manuscript. Otherwise, the reliability of the results may be decreased. Moreover, if some specific cutoffs were applied for the detections or predictions, they also need to be shown in the method section. For example, the criteria for searching the virulence-associated and antibiotic-associated genes listed at supplementary table 3 and 4 is not provided. The readers do not know how the authors run the software and how to obtain these results.

2. I would also suggest the authors to upload their scripts or commands for bioinformatic analyses online publicly. It can benefit the reproducibility and provide a guideline for the future applications as well.

3. From the *S. aureus* clinical isolates, several unknown MLSTs were found. It will be great if the authors can show more details about these unknown MLSTs. For example, the unknown MLST in figure 3 is quite close to ST-22, how different are they?

4. The statements at line 158-160 mentioned that "The genomes represent varied MLSTs. The 16 *S. haemolyticus* isolates examined here belonged to nine MLSTs, including a new genotype ST-74 (strain 51) assigned as a result of this study, and an isolate of unknown ST (strain 7A)". However, the data shown at figure 2 and figure S1 does not match with this statement. In both figure 2 and figure S1, ST-74 is for strain 7A and unknown ST is for strain 51. But the data in the supplementary table 2 does match the statement in the manuscript. The authors need to double check the data.

5. In the section of methods, line 292-294, the authors mentioned that "A total of 8 *S. aureus* and 14 *S. haemolyticus* consecutive non-duplicate isolates were collected from the Medical Microbiology Laboratory at Alexandria Main University Hospital (AMUH) between September and December 2015." However, the results and whole descriptions in the manuscript are based on 56 *S. aureus* and 10 *S. haemolyticus* isolates. Why the authors only use parts of the collection for the analyses. Perhaps, I miss it, but I did not find out the reason.

6. Based on geographical location, Middle East is the bridge between Asia and Europe. It will be interesting to see the phylogenomic analysis of the strains from Asia, Middle East and Europe. The spread and evolution of *Staphylococci* can be understood. If the authors can provide such information, the value of this study can be increased.

Reviewer #4 (Comments for the Author):

The article "Phylogenomic study of *Staphylococcus aureus* and *Staphylococcus haemolyticus* clinical isolates from Egypt" gives a distinct look at the two most pathogenic *Staphylococcus* species and types/clades of these species that are circulating in the Middle Eastern region, mainly Egypt. Additionally, the authors isolate 4 newly identified, distinct genotype strains based on their analyses. While this is certainly valuable information, particularly because the strains were acquired from active hospital infections, some of the importance and novelty of this study is lost in the type/clade jargon that is used without any indication of why these distinct types/clades are different or important to acknowledge. The manuscript could be improved by keeping a broader audience in mind who may be interested in phylogenetics or antibiotic resistance without the specific knowledge of jargon used in the *S. aureus* field in relation to strain types/clades. Two of the 4 figures in the manuscript refer to these types/clades, so a more thorough introduction and discussion would be very beneficial to comprehending the importance of the manuscript.

In this manuscript, *Staphylococcus aureus* and *Staphylococcus haemolyticus* clinical isolates from Egypt were collected. In total, 56 *S. aureus* and 10 *S. haemolyticus* isolates were collected and also performed WGS. *Staphylococci* are an important heterogeneous group of pathogens which can cause severe infection. The study of antibiotic resistance and virulence for these two strains can bring a large impact for the clinical research. However, only minority of *S. aureus* and *S. haemolyticus* studies provided the information from Middle East which is the connection between Asia and Europe. The authors of this manuscript cooperated Alexandria Main University Hospital to fill in this gap. All the WGS data including contigs, scaffolds and annotations are opened for public usages. Moreover, the antibiotic-associated and virulence-associated genes also provide useful information for the future clinical study. Furthermore, the phylogenomic study revealed the relationship between region and MLST as well.

Based on the manuscript, I have several suggestion and questions:

1. For the reproducibility, the parameters that the authors assigned for the software should be provided. If the default setting was used, it also needs to be written in the manuscript. Otherwise, the reliability of the results may be decreased. Moreover, if some specific cutoffs were applied for the detections or predictions, they also need to be shown in the method section. For example, the criteria for searching the virulence-associated and antibiotic-associated genes listed at supplementary table 3 and 4 is not provided. The readers do not know how the authors run the software and how to obtain these results.
2. I would also suggest the authors to upload their scripts or commands for bioinformatic analyses online publicly. It can benefit the reproducibility and provide a guideline for the future applications as well.
3. From the *S. aureus* clinical isolates, several unknown MLSTs were found. It will be great if the authors can show more details about these unknown MLSTs. For example, the unknown MLST in figure 3 is quite close to ST-22, how different are they?
4. The statements at line 158-160 mentioned that "The genomes represent varied MLSTs. The 16 *S. haemolyticus* isolates examined here belonged to nine MLSTs, including a new genotype ST-74 (strain 51) assigned as a result of this study, and an isolate of unknown ST (strain 7A)". However, the data shown at figure 2 and figure S1 does not match with this statement. In both figure2 and figure S1, ST-74 is for strain 7A and unknown ST is for strain 51. But the data in the supplementary table 2 does match the statement in the manuscript. The authors

need to double check the data.

5. In the section of methods, line 292-294, the authors mentioned that “A total of 8 *S. aureus* and 14 *S. haemolyticus* consecutive non-duplicate isolates were collected from the Medical Microbiology Laboratory at Alexandria Main University Hospital (AMUH) between September and December 2015.” However, the results and whole descriptions in the manuscript are based on 56 *S. aureus* and 10 *S. haemolyticus* isolates. Why the authors only use parts of the collection for the analyses. Perhaps, I miss it, but I did not find out the reason.
6. Based on geographical location, Middle East is the bridge between Asia and Europe. It will be interesting to see the phylogenomic analysis of the strains from Asia, Middle East and Europe. The spread and evolution of *Staphylococci* can be understood. If the authors can provide such information, the value of this study can be increased.

Response to Editorial & Reviewer Comments

Please find our responses below in bold. Line numbers corresponding to changes made in response to the reviewer comments correspond to line numbers in the resubmitted manuscript (without tracking) and are highlighted in green.

Editorial comments:

While I think the backbone of this paper is solid, it does need major revisions to make it understandable and digestible to a broader reading audience. The type/clade jargon could use some major explaining in the introduction and/or discussion to become understandable, and more statistical analyses should be done to support some of results.

We recognize that there is a lot of jargon surrounding *Staphylococcus*, including the multiple different methods for typing strains for global epidemiological studies. We have reviewed our manuscript and added to context of the typing jargon such that a broader audience can better understand and appreciate the results presented. Furthermore, we agree that it was difficult to interpret some of the results and discussion surrounding the clades. We have revised Figure 3 to include a visualization of the clades. We have also added this part to the introduction (Lines 96-99) “In that respect, WGS can be used to identify outbreak clones or clades, which are a group of independent isolates that share phenotypic and genotypic traits, most likely have a common ancestor, and form a branch on a phylogenetic tree (17–19)”.

As it currently stands, the paper is just a description study of sequences. Addition of relevant elements in the discussion would enhance the scientific impact. Please, also make sure that the findings support conclusions.

Due to the dearth of *S. aureus* and *S. haemolyticus* sequence data from the Arab region, the conclusions that can be drawn from our analysis have limitations. We have made this clear to the reader. Nevertheless, it is an important study for: (1) understanding these two important pathogens in the region, and (2) laying the foundation to future global studies that investigate the role that this region, and in particular Egypt, plays in the transmission of Staphylococci endemic to Asia and Staphylococci endemic to Europe.

Since the study involves bioinformatic analyses, the parameters (details) should be included in the manuscript to provide relevant information how the analyses were performed. Different parameter sets may influence the results and internal statistical tests significantly. I would strongly recommend providing such information or uploading it online for reproducibility.

We recognize that there may have been some confusion, noted by Reviewer #3, regarding the bioinformatic analyses. No new scripts were created. Within the revised manuscript, we have made this clear as well as a more thorough discussion of the parameters used for every step of our analysis. We have fully addressed these concerns in our response to Reviewer #3.

Reviewer #1 (Comments for the Author):

General comment
In their study, authors aimed the use of genomic data to investigate circulating strains of *S. aureus* and *S. haemolyticus* in Egypt. Despite the strength of genomic data provided, the scientific content, the structure as well as the wording of this article need to be significantly improved. English revision should be done for resubmission. Example: Line 74 (the infection it can cause range)

We have rephrased the specific statement indicated here.

“*S. aureus* is arguably the most clinically important staphylococcal species; it can be the cause of mild erythema to serious life-threatening ailments, including septicemia, pneumonia, and endocarditis” (Lines 74-76)

We also have reviewed the manuscript for wording.

Specific comments

The text of the Impact statement remains very descriptive and does not raise the impact of this study.

The biggest impact of our study is the simple fact that very few isolates have been sequenced from the Middle East. We included the 40 publicly available *S. aureus* and *S. haemolyticus* genomes available prior to our study in our comparative genomics study. We produced 56 *S. aureus* and 10 *S. haemolyticus* genomes as part of our study. This more than doubled the number of available sequenced isolates from the entire Middle East region. This is certainly an important contribution. The genomic investigation provides a more detailed view of the strains in circulation than traditional molecular typing strategies, which is the source of most of the current data in the Middle East. We have revised the impact statement to more clearly emphasize the significance of our work.

The Introduction should end at line 130. The following text describes the results.

We have made this change.

Line 124: Middle East and elsewhere. What does elsewhere mean? Africa and Mediterranean?

We have revised this statement as follows:

“Furthermore, Egypt’s cultural and geographical placement may facilitate local Staphylococcal exposure to international lineages, both from the Middle East as well as Asia, Europe, and Africa.” (Lines 130-131)

The results section begins with a paragraph without a subtitle which ultimately describes more of a methodology and very few of the results.

The production of these genomes, which essentially doubles the number of publicly available genomes from Middle Eastern isolates for these two species, is a result. However, we recognize that the value of these genome sequences is through the comparative analysis. We have restructured the start of the Results such that the first subheading is “Strain genotyping” where we introduce the new 66 strains and their genomes. Moreover, we have focused the introduction of these results. (Lines 142-148)

The discussion, which should be a discussion of the results, relates more elements that have their place in the introduction. The discussion is not hard-hitting and should be better developed.

Thank you for this comment. We have revised the discussion accordingly.

In Methods (Bacterial isolates), the authors speak of 89 *S. aureus* and 14 *S. Haemolyticus*, which does not correspond to the figures described in the abstract.

Thank you for noting this discrepancy. We have corrected the numbers reported in the methods:

“Fifty-six *S. aureus* and 10 *S. haemolyticus* consecutive non-duplicate isolates were collected from the Medical Microbiology Laboratory at Alexandria Main University Hospital (AMUH) between September and December 2015.” (Lines 302-304)

In this same paragraph, the authors describe poorly performing conventional methods for the identification of the genus *Staphylococcus* identification. I would have expected confirmation based on the genomic data in their possession.

Initially, the isolates’ species were confirmed using conventional methods. These initial taxonomic classifications were later confirmed using the genomic data. We have made this fact clearer in the methods (Lines 336-339). checkM was run to assess completeness and contamination of the assemblies. This process was

conducted through PATRIC and the user is required to specify the genus. It's important to note that PATRIC by default ascertains if this is the right genus for comparison. The results of this first step in genome assembly evaluation confirmed our initial taxonomic designation. It was further confirmed upon submission to NCBI and PGAP annotation, which by default runs analyses to confirm the user-supplied genus and species.

The low number of non-Egyptian genomes in this study cannot allow a prevalence comparison among MLST types. Authors should use this genomic data with another angle.

We agree with the reviewer, the number of genomes from the Middle East is scant in comparison with, e.g., Europe and the USA. With the limited publicly available data, of which our contribution here more than doubled this resource, it is not possible at this time to speak of prevalence. However, our study is a very important first step in beginning to explore the diversity of lineages within the Middle East, and more specifically Egypt. We believe it is important that this is emphasized throughout the scientific literature; a global perspective of pathogen covalence is desperately needed.

Reviewer #2 (Comments for the Author):

The research article proposed by Montelongo et al entitled "Phylogenomic study of Staphylococcus aureus and Staphylococcus haemolyticus clinical isolates from Egypt" describes one of the important topics. In general the article is of great impact to readers working in the healthcare settings, especially as considering it performing the WGS for large number of isolates, however, some minor issues need to be addressed first before considering final publication. Those are shown below.
1- The author need to add in the introduction section a short paragraph about the clinical outcomes and the PVL positive S. aureus specially in Egypt.

We have included in the introduction information about the clinical outcomes of PVL.

“PVL-positive strains are most frequently associated with skin and soft-tissue disease, although incidences have been associated with pneumonia and bacteremia.” (Lines 113-114)

We have expanded our results to include more details about PVL strains. (Lines 191-196)

Specific discussions of PVL-positive strains in Egypt are limited. We have included these in the Discussion. (Lines 260-263)

2- PVL positive isolates are mainly accompanied with severe outcomes like the brain abscess so illustrating such correlation will raise the significance of the work.

A paragraph was added to the introduction to stress the significance and prevalence of PVL-positive isolates in Egypt.

See response to the prior comment.

3- The authors need to emphasize that the geographical location of Egypt plays an important role in the dissemination of various lineages and types within this country.

First, we have emphasized this point in the introduction:

“Because of its central location as well as its political and historical role, Egypt presents a unique case-study for staphylococcal distribution and exchange in the Arab region (33). Furthermore, Egypt’s cultural and geographical placement may facilitate local Staphylococcal exposure to international lineages from the Middle East, as well as Asia, Europe, and Africa.” (Lines 130-131)

We have emphasized this again in the discussion through our discussion of sequence types (STs) that are prevalent in Europe and Asia and are found in Egypt. (Lines 236-246)

4- Please rewrite the part of the open pangenome to be simpler and clearer to the readers.

We made revisions in the presentation of the pangenome results (beginning Line 174) as well as our discussion of this analysis (Line 254). We have also added detail to the methods regarding how these computations were performed (beginning Line 363).

5- Table 2: Are these all the identified list of the virulence factors?? too little.

These are just the virulence factors within the core genome, i.e., encoded by all strains. Supplementary Table S3 lists all the virulence factors.

Reviewer #3 (Comments for the Author):

In this manuscript, Staphylococcus aureus and Staphylococcus haemolyticus clinical isolates from Egypt were collected. In total, 56 S. aureus and 10 S. haemolyticus isolates were collected and also performed WGS. Staphylococci are an important

heterogenous group of pathogens which can cause severe infection. The study of antibiotic resistance and virulence for these two strains can bring a large impact for the clinical research. However, only minority of *S. aureus* and *S. haemolyticus* studies provided the information from Middle East which is the connection between Asia and Europe. The authors of this manuscript cooperated Alexandria Main University Hospital to fill in this gap. All the WGS data including contigs, scaffolds and annotations are opened for public usages. Moreover, the antibiotic-associated and virulence-associated genes also provide useful information for the future clinical study. Furthermore, the phylogenomic study revealed the relationship between region and MLST as well.

Based on the manuscript, I have several suggestion and questions:

1. For the reproducibility, the parameters that the authors assigned for the software should be provided. If the default setting was used, it also needs to be written in the manuscript. Otherwise, the reliability of the results may be decreased. Moreover, if some specific cutoffs were applied for the detections or predictions, they also need to be shown in the method section. For example, the criteria for searching the virulence-associated and antibiotic-associated genes listed at supplementary table 3 and 4 is not provided. The readers do not know how the authors run the software and how to obtain these results.

The parameters used for raw read processing and assembly were included in our prior manuscript (beginning Line 330). We neglected to mention that completeness and contamination were assessed via PATRIC during the PATRIC annotation process. We have included this information (Lines 338-339).

We have explicitly listed the parameters (if there were any) for the MLST, SpaTyper, SCCmecFinder, and VFAnalyzer analyses (beginning Line 350).

2. I would also suggest the authors to upload their scripts or commands for bioinformatic analyses online publicly. It can benefit the reproducibility and provide a guideline for the future applications as well.

No new scripts were developed as part of this work. We recognize that this may have been misleading when we referred to the *anvi'o* "scripts". These are in fact functions that can be called through the *anvi'o* environment. For the identification of the single-copy core genome we did specify that only gene clusters that were conserved among all genomes and occur once per genome were selected; we have now explicitly listed these parameters (beginning Line 363). Unless noted, default parameters for these *anvi'o* functions were used; *anvi'o* offers fantastic documentation to support users unfamiliar with these functions. For phylogenomic and phylogenetic tree derivation and visualization, default parameters were used, and we have explicitly stated this in the text (Line 371-375).

3. From the *S. aureus* clinical isolates, several unknown MLSTs were found. It will be great if the authors can show more details about these unknown MLSTs. For example, the unknown MLST in figure 3 is quite close to ST-22, how different are they?

This statement was added to the text: “AA32 however closely resembled ST-22 encountered among 7 strains, showing different alleles among 2/7 housekeeping genes constituting the MLST typing scheme of *S. aureus*.” (Lines 158-160)

4. The statements at line 158-160 mentioned that "The genomes represent varied MLSTs. The 16 *S. haemolyticus* isolates examined here belonged to nine MLSTs, including a new genotype ST-74 (strain 51) assigned as a result of this study, and an isolate of unknown ST (strain 7A)". However, the data shown at figure 2 and figure S1 does not match with this statement. In both figure2 and figure S1, ST-74 is for strain 7A and unknown ST is for strain 51. But the data in the supplementary table 2 does match the statement in the manuscript. The authors need to double check the data.

The authors thank the reviewer for noting this; both figures have been corrected.

5. In the section of methods, line 292-294, the authors mentioned that "A total of 8 *S. aureus* and 14 *S. haemolyticus* consecutive non-duplicate isolates were collected from the Medical Microbiology Laboratory at Alexandria Main University Hospital (AMUH) between September and December 2015." However, the results and whole descriptions in the manuscript are based on 56 *S. aureus* and 10 *S. haemolyticus* isolates. Why the authors only use parts of the collection for the analyses. Perhaps, I miss it, but I did not find out the reason.

Thank you for noting this discrepancy. We have corrected the numbers reported in the methods:

“Fifty-six *S. aureus* and 10 *S. haemolyticus* consecutive non-duplicate isolates were collected from the Medical Microbiology Laboratory at Alexandria Main University Hospital (AMUH) between September and December 2015.” (Lines 302-304)

6. Based on geographical location, Middle East is the bridge between Asia and Europe. It will be interesting to see the phylogenomic analysis of the strains from Asia, Middle East and Europe. The spread and evolution of Staphylococci can be understood. If the authors can provide such information, the value of this study can be increased.

We certainly agree that such an examination would be quite interesting. As our study highlights, there is very little information about strains from the Middle East. This is most evident for *S. haemolyticus*, in which very few strains have been sequenced, and all are from Egypt. We have made a conscious effort to

bring this dearth of data to the reader's attention. First, we have emphasized this point in the introduction:

“Because of its central location as well as its political and historical role, Egypt presents a unique case-study for staphylococcal distribution and exchange in the Arab region (33). Furthermore, Egypt’s cultural and geographical placement may facilitate local Staphylococcal exposure to international lineages from the Middle East, as well as Asia, Europe, and Africa.” (Lines 130-131)

We also have provided what little analysis is possible looking at dominant sequence types (STs) in the regions. We have included in the discussion of sequence types (STs) that are prevalent in Europe and Asia and are found in Egypt. (Lines 236-246)

Reviewer #4 (Comments for the Author):

The article "Phylogenomic study of Staphylococcus aureus and Staphylococcus haemolyticus clinical isolates from Egypt" gives a distinct look at the two most pathogenic Staphylococcus species and types/clades of these species that are circulating in the Middle Eastern region, mainly Egypt. Additionally, the authors isolate 4 newly identified, distinct genotype strains based on their analyses. While this is certainly valuable information, particularly because the strains were acquired from active hospital infections, some of the importance and novelty of this study is lost in the type/clade jargon that is used without any indication of why these distinct types/clades are different or important to acknowledge. The manuscript could be improved by keeping a broader audience in mind who may be interested in phylogenetics or antibiotic resistance without the specific knowledge of jargon used in the S. aureus field in relation to strain types/clades. Two of the 4 figures in the manuscript refer to these types/clades, so a more thorough introduction and discussion would be very beneficial to comprehending the importance of the manuscript.

We recognize that our prior manuscript made it difficult to ascertain which strains belonged to which clade. We have revised Fig. 3 such that the reader can easily identify the clades within the tree. This enables easier interpretation of the discussion of these clades. We have also added this part to the introduction (Lines 96-99) “In that respect, WGS can be used to identify outbreak clones or clades, which are a group of independent isolates that share phenotypic and genotypic traits, most likely have a common ancestor, and form a branch on a phylogenetic tree (17–19)”.

April 24, 2022

Dr. Alaa Abouelfetouh
Faculty of Pharmacy, Alexandria University and Alamein International University
Microbiology and Immunology
1 Khartoum Square, Azarita
Alexandria 21521
Egypt

Re: Spectrum02413-21R1-A (Phylogenomic study of *Staphylococcus aureus* and *Staphylococcus haemolyticus* clinical isolates from Egypt)

Dear Dr. Alaa Abouelfetouh:

Thank you for submitting your manuscript to Microbiology Spectrum. When submitting the revised version of your paper, please provide (1) point-by-point responses to the issues raised by the editor as file type "Response to Reviewers," not in your cover letter, and (2) a PDF file that indicates the changes from the original submission (by highlighting or underlining the changes) as file type "Marked Up Manuscript - For Review Only". Please use this link to submit your revised manuscript - we strongly recommend that you submit your paper within the next 60 days or reach out to me. Detailed instructions on submitting your revised paper are below.

Link Not Available

Sincerely,

Hermine Mkrtchyan

Journals Department
Editor comments:

Although it has previously been suggested (Reviewer 1) that 'English revision must be done for resubmission', I still feel English has not been improved.

The entire manuscript must be edited by a native English speaker but who also is an expert in the field.

The manuscript still does not read smoothly. The sentences and paragraphs are sometimes disconnected. There are differences in tenses, e.g. lines 228-229; 241-243.

Below are a few examples, but full editing is required is throughout the manuscript. Please, note this is the last chance to revise the manuscript for acceptable standards.

Title

Replace 'Phylogenomic study'. Perhaps better to say 'Whole genome sequencing of'

Line 38: Replace 'resistant infections' with 'antimicrobial resistant pathogens causing infections' or something similar.

Please, rephrase sentences writing them in an academic/scientific style throughout the manuscript narrowing down to Egypt as a main location for your study, e.g.

Line 39: 'Little is known about the population structure of drug resistant staphylococci recovered from patients and clinical settings in Egypt'.

Please rename 'staphylococcal Middle Eastern isolates'.

Lines 45-47: Clarify 'S. aureus multilocus sequence typing (MLST) types'. Perhaps multilocus sequence types (MLST)?

Lines: 45-46: 'including 3 and 1 new MLSTs' - start a new sentence and say which of the species possessed 3 and which one 1 new STs.

Line 56: Narrow it down to Egypt.

Line 65: 'four new MLSTs were identified among the Middle East strains' - my understanding was that isolates recovered from Alexandra hospital had new sequence types, did they not?

Lines 64-69: Rephrase the revised sections in the Impact statement.

Please, revise the Introduction section to improve throughout.

E.g. the sentence below can easily be revised to make it easy for the reader to understand the challenges associated with S. aureus treatment.

Lines 82-84: 'A difficulty in treating and controlling S. aureus stems from its prevalence and increasing resistance to clinically used antibiotics, resulting in it being one of the leading agents for nosocomial and community-acquired infections'.

Please, clarify the following sentence:

Lines 84-87: 'S. haemolyticus is the second most common staphylococcal species isolated in human blood cultures and a prominent reservoir for antibiotic resistance genes, which can be shared with other Staphylococci, including S. aureus'

Staphylococci - must be written with small s.

Line 94: 'tracking a broad range of clones over a global area' - please clarify 'over a global area'

Line 146 - 'Arab region 'appears here suddenly.

Review lines 88-111.

Rephrase or remove lines 107-110.

Lines 395-396: Parameters for the MALDI TOF must be provided.

Line 393: Mannitol Salt Agar (MSA) - manufacturer, city, country ?

Line 392: 'The identity of the isolates was determined' - All isolates were identified using

Table 1: fifth row: 'No mecA detected' - all the others are with small n.

Previously, Reviewer 3 suggested to add phylogenetic analysis of strains from Asia, Middle East and Europe. This is possible using already archived data that can be obtained from the ENA or NCBI.

Previously, Reviewer 2 suggested to add about PVL in Introduction, however the addition is not informative to emphasise the importance of PVL.

Staff Comments:

Preparing Revision Guidelines

Please return the manuscript within 60 days; if you cannot complete the modification within this time period, please contact me. If you do not wish to modify the manuscript and prefer to submit it to another journal, please notify me of your decision immediately so that the manuscript may be formally withdrawn from consideration by Microbiology Spectrum.

We have attempted to respond positively to each of the editor's concerns. We hope that these responses are satisfactory.

Although it has previously been suggested (Reviewer 1) that 'English revision must be done for resubmission', I still feel English has not been improved.

The entire manuscript must be edited by a native English speaker but who also is an expert in the field.

The manuscript still does not read smoothly. The sentences and paragraphs are sometimes disconnected. There are differences in tenses, e.g. lines 228-229; 241-243.

Response: *The native English-speaking authors have reviewed and edited the manuscript with the editor's concerns in mind. Extensive edits are highlighted in yellow.*

Below are a few examples, but full editing is required is throughout the manuscript. Please, note this is the last chance to revise the manuscript for acceptable standards.

Title

Replace 'Phylogenomic study'. Perhaps better to say 'Whole genome sequencing of ...'

Response: *Changed as requested (line 1).*

Line 38: Replace 'resistant infections' with 'antimicrobial resistant pathogens causing infections' or something similar.

Response: *Changed as requested (line 35). It now reads as follows:*

"Infections caused by antibiotic resistant Staphylococcus are a global concern."

Please, rephrase sentences writing them in an academic/scientific style throughout the manuscript narrowing down to Egypt as a main location for your study, e.g.

Line 39: 'Little is known about the population structure of drug resistant staphylococci recovered from patients and clinical settings in Egypt'.

Response: *Changed as requested (lines 38-39). It now reads as follows:*

"...the population structure of antibiotic resistant staphylococci recovered from patients and clinical settings in Egypt remains uncharacterized."

Please rename 'staphylococcal Middle Eastern isolates'.

Response: *Changed as requested (line 44). It now reads as follows:*

"... staphylococcal isolates from the Middle East."

Elsewhere, we use the same approach.

Lines 45-47: Clarify 'S. aureus multilocus sequence typing (MLST) types'. Perhaps multilocus sequence types (MLST)?

Response: *Changed as requested (line 45).*

Lines: 45-46: 'including 3 and 1 new MLSTs' - start a new sentence and say which of the species possessed 3 and which one 1 new STs.

Response: *Changed as requested (lines 45-46).*

“These genomes include 20 S. aureus multilocus sequence types (MLST), including 3 new ones. They also include 9 S. haemolyticus MLSTs, including 1 new one.”

Line 56: Narrow it down to Egypt.

Response: *Changed as requested (line 55).*

Line 65: 'four new MLSTs were identified among the Middle East strains' - my understanding was that isolates recovered from Alexandra hospital had new sequence types, did they not?

Response: *No, the newly identified MLSTs were from previously deposited assemblies.*

However, the text has been altered to remove any confusion; the “4 new MLSTs” statement now comes after we make it clear that our analysis also included publicly available genomes.

Lines 64-69: Rephrase the revised sections in the Impact statement.

Response: *Rephrased, as requested (lines 61-64). It now reads as follows:*

“For example, we identified 4 new MLSTs. Most strains harbored genes associated with multidrug resistance, toxin production, biofilm formation, and immune evasion. These data provide invaluable insight for future antibiotic stewardship and infection control within the Middle East.”

Please, revise the Introduction section to improve throughout.

Response: *Done. We made numerous changes to make it read better.*

E.g. the sentence below can easily be revised to make it easy for the reader to understand the challenges associated with S. aureus treatment.

Lines 82-84: 'A difficulty in treating and controlling S. aureus stems from its prevalence and increasing resistance to clinically used antibiotics, resulting in it being one of the leading agents for nosocomial and community-acquired infections'.

Response: *Rephrased, as requested (lines 74-77). It now reads as follows:*

“A challenge in treating and controlling S. aureus stems from both its prevalence and its increasing resistance to clinically used antibiotics. Together, they make S. aureus one of the leading agents of nosocomial and community-acquired infections (3, 4).”

Please, clarify the following sentence:

Lines 84-87: 'S. haemolyticus is the second most common staphylococcal species isolated in human blood cultures and a prominent reservoir for antibiotic resistance genes, which can be shared with other Staphylococci, including S. aureus'

Response: *Clarified, as requested (lines 77-79). It now reads as follows:*

“S. haemolyticus is the second most common staphylococcal species isolated from human blood culture. It can be a reservoir for antibiotic resistance genes, which can be shared with other staphylococci, including S. aureus (5–7).”

Staphylococci - must be written with small s.

Response: *Correct. Fixed.*

Line 94: 'tracking a broad range of clones over a global area' - please clarify 'over a global area'

Response: *Deleted global, as it is unnecessary (line 82).*

Line 146 - 'Arab region' appears here suddenly.

Response: *Changed to Middle East throughout.*

Review lines 88-111.

Response: *Done.*

Rephrase or remove lines 107-110.

Response: *Rephrased.*

Lines 395-396: Parameters for the MALDI TOF must be provided.

Response: *Software now provided (line 324).*

Line 393: Mannitol Salt Agar (MSA) - manufacturer, city, country?

Response: *Lines 321-322. Mannitol Salt Agar (MSA) (Oxoid Ltd, England).*

Line 392: 'The identity of the isolates was determined' - All isolates were identified using

Response: *Rephrased, as directed (line 320).*

Table 1: fifth row: 'No mecA detected' - all the others are with small n.

Response: *Fixed.*

Previously, Reviewer 3 suggested to add phylogenetic analysis of strains from Asia, Middle East and Europe. This is possible using already archived data that can be obtained from the ENA or NCBI.

Response: *As suggested, we have added this analysis (lines 220-227).*

“Next, we compared the Middle Eastern S. aureus and S. haemolyticus isolate genomes to genomes of isolates collected from Europe and Asia. We restricted our analysis to isolates collected from 2010 through 2019, as the Egyptian isolates sequenced in this study were collected in 2015. In total, 302 S. aureus and 82 S. haemolyticus genomes were included in this analysis (see Methods). The core genome was computed for both species, identifying a single-copy core genome of 445 genes and 1,071 genes for S. aureus and S. haemolyticus, respectively.

Based upon these core genomes, the phylogenies were derived, indicating the continent of origin for each genome (Fig. 5)."

Also, the methods (lines 392-406):

*"Intercontinental comparisons of *S. aureus* and *S. haemolyticus* genomes were also conducted. All publicly available genomes for these two species were retrieved from NCBI. Based upon the metadata associated with each genome, we restricted our analysis to strains collected between 2010 and 2019 and from a country/region in Asia, Europe, or the Middle East. The collection date range was implemented given that the Egypt isolates were collected in 2015. All publicly available *S. haemolyticus* genomes meeting these criteria were included in the analysis as well as the six previously deposited *S. haemolyticus* genomes from Egypt for which no collection date was available. This dataset thus includes 82 genomes. In total, 1,790 *S. aureus* genomes met the date and country/region criteria. These genomes were subsampled such that, for isolates from Asia and Europe, three genomes were randomly selected for each year and country/region combination; all isolates from the Middle East were included. In total, 302 genomes were selected for analysis. **Supplementary Table S5** lists details about the genomes included for both species. For the intercontinental *S. aureus* and intercontinental *S. haemolyticus* genomes, the core genome was identified via *anvi'o*, and a phylogenetic tree was derived via *FastTree* and visualized via *iTOL* as described above."*

Previously, Reviewer 2 suggested to add about PVL in Introduction, however the addition is not informative to emphasise the importance of PVL.

Response: *Added a bit more information (lines 86-91, especially 89-90).*

*"Lastly, *S. aureus* strains are often assayed for the virulence factor Panton-Valentine leucocidin (PVL), which is common among community-acquired MRSA (CA-MRSA) strains and rare among hospital-associated MRSA (HA-MRSA) strains (11). PVL is thought to contribute to epidemic spread (13), and many MRSA strains in circulation in the United States and Europe, e.g., USA300 strains, are PVL-positive (14)."*

May 31, 2022

Dr. Alaa Abouelfetouh
Faculty of Pharmacy, Alexandria University and Alamein International University
Microbiology and Immunology
1 Khartoum Square, Azarita
Alexandria 21521
Egypt

Re: Spectrum02413-21R2 (Whole genome sequencing of *Staphylococcus aureus* and *Staphylococcus haemolyticus* clinical isolates from Egypt)

Dear Dr. Alaa Abouelfetouh:

Your manuscript has been accepted, and I am forwarding it to the ASM Journals Department for publication. You will be notified when your proofs are ready to be viewed.

When you receive the proof, I would appreciate if you could rephrase the following 'Figure 1. Genome analysis of 90 Arab *S. aureus* strains'. Perhaps it would be better to replace '90 Arab *S. aureus* strains' in the sentence below with '90 *S. aureus* isolates recovered from Middle East' or something similar?

Sincerely,

Hermine Mkrtchyan
Editor, Microbiology Spectrum

Journals Department
Supplemental Material: Accept
Supplemental Material: Accept
Supplemental Table 5: Accept